

# Degreasing and bleaching bones using light sources as a tool to increase the safety of teaching osteology at the University of Veterinary Sciences Brno

Ondřej Horák[1], Martin Pyszko[1], Václav Páral[1] and Ondřej Šandor[2]

[1] Department of Anatomy, Histology and Embryology/Faculty of Veterinary Medicine, University of Veterinary Sciences Brno, Brno, Czech Republic
[2] Department of Pathology and Parasitology/Faculty of Veterinary Medicine, University of Veterinary Sciences Brno, Brno, Czech Republic

## ABSTRACT

The key part of creating bone material for teaching is degreasing and whitening it. However, the substances used are often dangerous and toxic. We tested and compared safer methods based on two physical variables. These are light and heat. The material for our study was 45 femurs from 23 adult domestic dogs (*Canis lupus f. familiaris*). The bones were divided into three groups of 15 pieces according to the method used to remove muscles and ligaments from their surface. Five femurs from each group were exposed to three different light sources for 28 days—sunlight, warm light from a classical incandescent light bulb and cold light by a LED bulb. At regular intervals, the change in the colour of the bone surface and the amount of fat loss from the medullary cavity was also monitored. The best degreasing and bleaching results were achieved in macerated bones exposed to sunlight. They achieved the required condition as early as 21 days after the start of sun exposure. The biggest problem was haemoglobin, which permeated through the Haversian canals and discoloured the bone tissue. The results showed that the use of light and heat is a suitable and safe alternative to chemical methods of degreasing and bleaching bones. The disadvantage is the length of time, especially for native material.

## INTRODUCTION

Bone material from real animals is, even in today's age of artificial models and computer simulations, an irreplaceable teaching aid in practical anatomy exercises at the University of Veterinary Sciences Brno. There are many methods of obtaining clean and white bones from the bodies of dead animals (*Hefti et al., 1980*; *Simonsen et al., 2011*; *Uhre et al., 2015*; *Husch et al., 2021*). However, after the removal of muscles and ligaments the bones must be degreased and whitened (*Fages et al., 1994*; *Hussain et al., 2007*). Degreasing and subsequent bleaching are key steps that determine the overall success or failure of bone material creation (*Von Endt, Ross & Hare, 1999*). Organic solvents such as gasoline, tetrachloride, xylene, acetone and alcohols are most often used to degrease bones, *i.e.*, to

Corresponding author
Martin Pyszko, pyszkom@vfu.cz

remove fat from bone marrow (*Mahon, Maboke & Myburgh, 2021*). Unfortunately, many of these substances also have side effects such as flammability, toxicity, carcinogenicity and teratogenicity (*Recknagel et al., 1989*). Certain substances used for final bone bleaching, such as peroxides and chlorine compounds, share the same side effects as organic solvents (*Winterbourn, 1995*; *Mairs, Swift & Rutty, 2004*). Chemical residues that are left on material after degreasing or bleaching must be removed completely, unfortunately it is not always possible. That is why we decided to test and compare safer methods based on two physical quantities—light and heat.

## MATERIALS AND METHODS

The material for our study was 45 femurs from 23 adult domestic dogs (*Canis lupus f. familiaris*). The dogs were not preserved in formalin (10% formaldehyde), but they were kept in a freezer (−18 °C) for several months. After a sufficient number of dogs were reached, these dogs were defrosted and the femurs were dissected from their pelvic limbs. Breed and sex were not monitored. Our only condition was approximate length of femurs, which was 15–20 cm. No permits were required for this study because the analyses were performed exclusively on the bones of dead animals that were the property of our department. None of the dogs were euthanized for the purpose of our study and did not suffer from musculoskeletal disorders. No additional samples were collected or used for this study. The bones were divided into three groups of 15 pieces. Each group contained bones cleaned by a different preparation method. The first group was biologically macerated in an aqueous solution with rotting bacteria. The duration of maceration was 2 months under laboratory conditions at a temperature of 20–25 °C. The bones of the second group were boiled in water without any admixture for 3 h and then cleaned with a stream of water and a toothbrush. The last group was used in the native state, the muscles and ligaments were removed manually during the autopsy.

The first method used to degrease and bleach the above-described material was exposure to summer sunlight. We used 15 bones (five macerated, five cooked and five native). The experiment was done in one of the hottest months of the year (August) to ensure the strongest possible light and heat. The light-day length for 2021 at the experiment site (49.2° north latitude and 16.6° east longitude) was between 13:35 and 15:15 h. The average daily temperature was 18.5 °C (in the shade), with a maximum of 25–35 °C in the afternoon. The femurs were put on a 5 cm high layer of sand and placed on the flat roof of the building to minimize the influence of shade. There were no trees or other buildings in the area. The bones were left for 28 days and were regularly checked and photographed. Photo documentation was done on day zero and then again on day 1, 3, 7, 14, 21 and 28. The second and third methods were very similar to the first but differed in the type and strength of the light source used. The actual execution of the experiment in these two methods was in the laboratory under precisely defined conditions. The temperature in laboratory was 20 °C all the time for both groups. In the second method, a classical incandescent light bulb ("warm light") of 100 W and 1340 lumen (TESCO) luminosity was used as a source of light and heat. In the third method, we used the most environmentally friendly option, which was a 14 W LED bulb ("cold light") and

**Scheme 1 Colour scheme for bone bleaching.** RGB—colour model (red, green, blue), HEX—hexadecimal colour model, Lab—CIELab colour mode, based on the human perception of colour.

1521 lumen (EMOS) luminosity. The bones were placed on a 5 cm high layer of sand in cuvettes 55 × 30 × 30 cm (height × width × depth). The distance between the two light sources and group of experimental bones was 25 cm. The bones were exposed to light for 14 h a day (from 7:00 to 21:00) for 28 days. After 21:00, the bulbs were switched off to "simulate" night darkness and cooling down to imitate the conditions of method one as much as possible. The bones were placed in an absolutely dark room without windows (no remaining light level). For both the second and third method, again 15 bones (five macerated, five boiled and five native) were used as for method one. These bones were also photographed on days 0, 1, 3, 7, 14, 21 and 28. The amount of fat loss from the medullary cavity was determined by subjective assessment by one observer from zero to three crosses. Zero cross was assigned to the bones without any leakage of fat. One cross was assigned to the bones with slight leakage of fat into the surrounding substrate (sand) at up to 5 mm from the bone. Two crosses were assigned to bones with leakage of fat into the surrounding substrate between 5–10 mm from the bone. Three crosses were assigned to bones with fat leaking further than 10 mm from the bone. The degree of whiteness of the exposed bones was determined using a scale from 1 to 10, which we have created for our needs (Scheme 1). A number 1 was given to dark dirty bone and a number 10 to white bone bleached by chemical methods. We could not use the available whiteness scales, used by dentists, because most of the bones were darker than the darkest degree of these scales at the beginning of the experiment.

## RESULTS

The results of our experiment showed that the rate of degreasing and bleaching of bones using different light sources is significantly divergent (Tables 1 and 2). The best degreasing was achieved in boiled and native bones. Macerated bones released fat from the bone

**Table 1 Degree of bone degreasing in relation to time.**

| Degreasing method | Group of bones | Day 0 | Day 1 | Day 3 | Day 7 | Day 14 | Day 21 | Day 28 |
|---|---|---|---|---|---|---|---|---|
| Sun light | A | – | + | + | + | + | + | – |
| | B | – | + | + | ++ | ++ | + | + |
| | C | – | + | + | ++ | +++ | ++ | + |
| Warm light | A | – | – | + | + | + | – | – |
| | B | – | – | + | ++ | ++ | + | – |
| | C | – | – | + | ++ | +++ | ++ | + |
| Cold light | A | – | – | – | – | – | – | – |
| | B | – | – | – | – | – | – | – |
| | C | – | – | – | – | + | – | – |

Note:
A—macerated bone, B—boiled bone, C—natural bone, zero cross (–) = 0 mm, one cross (+) = 0–5 mm, two crosses (++) = 5–10 mm and three crosses (+++) = more than 10 mm.

**Table 2 Degree of bone bleaching in relation to time.**

| Bleaching method | Group of bones | Day 0 | Day 1 | Day 3 | Day 7 | Day 14 | Day 21 | Day 28 |
|---|---|---|---|---|---|---|---|---|
| Sun light | A | 1 | 2 | 2 | 5 | 7 | 8 | 9 |
| | B | 2 | 3 | 3 | 4 | 5 | 6 | 7 |
| | C | 1 | 2 | 2 | 2 | 3 | 4 | 4 |
| Warm light | A | 2 | 2 | 3 | 5 | 5 | 5 | 5 |
| | B | 2 | 2 | 3 | 5 | 5 | 6 | 6 |
| | C | 2 | 2 | 2 | 3 | 3 | 3 | 3 |
| Cold light | A | 2 | 2 | 2 | 3 | 4 | 4 | 4 |
| | B | 2 | 2 | 2 | 3 | 4 | 4 | 4 |
| | C | 2 | 2 | 2 | 2 | 2 | 2 | 2 |

Note:
A—macerated bone, B—boiled bone, C—natural bone, 1–9 according to Scheme 1, smaller number = darker colour of bone, higher number = lighter colour of bone.

marrow only slightly and at most from their epiphyses, where the thickness of the bone compact is the smallest. The fastest release of fat from the bones was found in sunlight-exposed bones, followed by the warm light-exposed group. The slowest fat leakage was from the third method (exposure to cold light). Based on these results, it is evident that the key parameter for the perfect degreasing of treated bones is temperature and not light (Table 1). Bones placed in warmer environments degrease faster and better than bones placed in cooler environments.

The best and fastest bone bleaching was found in sun-exposed macerated bones (Table 2 and Fig. 1). The worst result of bone bleaching was in native bones (Fig. 2). In these cases, red haemoglobin from the bone marrow entered the Haversian canal system in the bone tissue that were placed in the sun. This resulted in a non-physiological reddish discoloration of the bone. The boiled bones did not whiten completely (did not have colour grade 9 or 10 from Scheme 1) for the duration of our experiment. Their final colouring after 28 days corresponded most closely to the physiological colour of living bone tissue (colour grade 7 from Scheme 1).

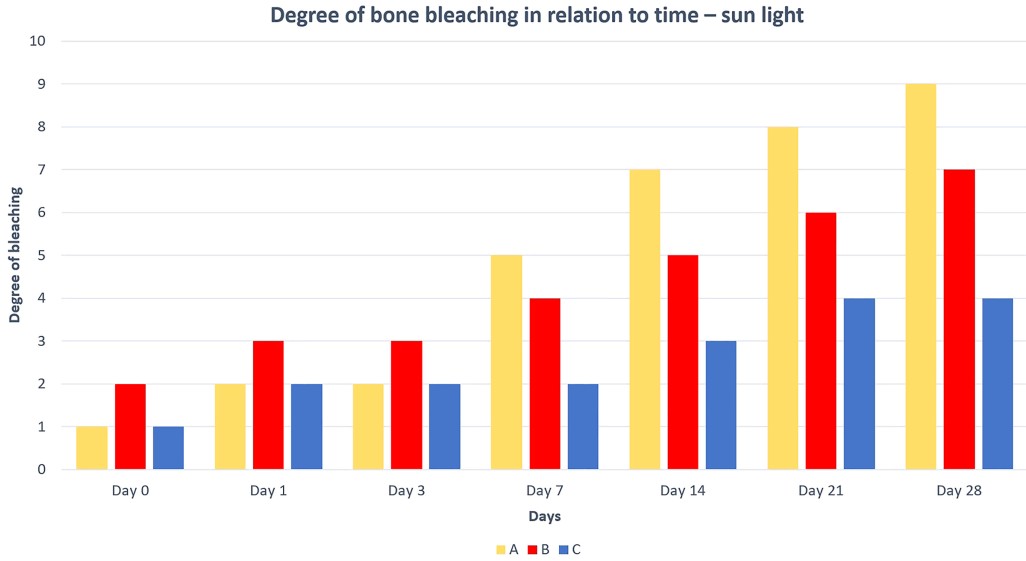

**Figure 1** **Degree of bone bleaching in relation to time—sunlight.** A—macerated bone, B—boiled bone, C—natural bone, x-axis = days, y-axis = degree of bleaching.

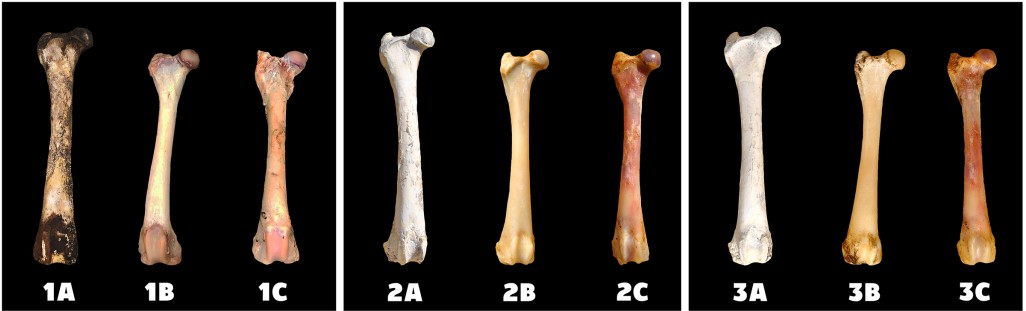

**Figure 2** **Changes at the right femur of three dogs exposed to direct sunlight.** A—macerated bone, B—boiled bone, C—natural bone, 1A to 1C—status at the start of the experiment, 2A to 2C—status after 14 days of exposure, 3A to 3C—status at the end of the experiment.

Differences in the speed and intensity of bone whitening by "artificial" warm light (Fig. 3) and cold light (Fig. 4) were negligible compared to the first method. In both cases, exposure was insufficient for 28 days as there was still a non-physiological red or brown colour at the epiphysis site due to fat and connective tissue remnants (Figs. 5 and 6).

## DISCUSSION

Most of the chemicals commonly used for degreasing and bleaching bones are flammable and explosive, toxic or have negative effects on human health (*Recknagel et al., 1989*; *Winterbourn, 1995*; *Mann & Berryman, 2012*). Even the relatively safe tetrachloromethane (carbon tetrachloride), which is non-flammable and non-explosive, cannot be recommended in other than laboratory conditions. When tetrachloromethane is heated and in contact with oxygen, phosgene is produced and chlorine gas is released into the air
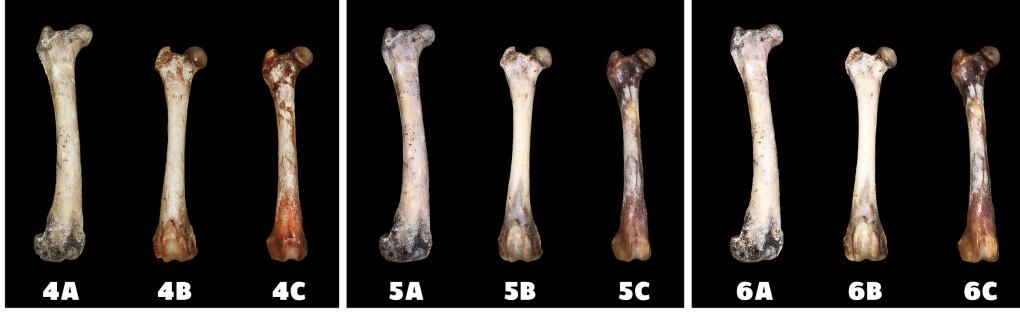

**Figure 3 Changes at the right femur of three dogs exposed to a "warm light" light bulb.** A—macerated bone, B—boiled bone, C—natural bone, 4A to 4C —status at the start of the experiment, 5A to 5C—status after 14 exposure days, 6A to 6C—status at the end of the experiment.

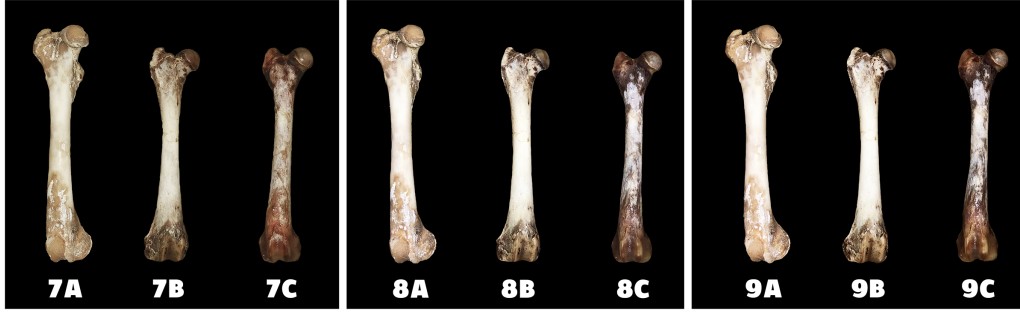

**Figure 4 Changes at the right femur of three dogs exposed to a "cold light" light bulb.** A—macerated bone, B—boiled bone, C—natural bone, 7A to 7C—status at the start of the experiment, 8A to 8C—status after 14 exposure days, 9A to 9C—status at the end of the experiment.

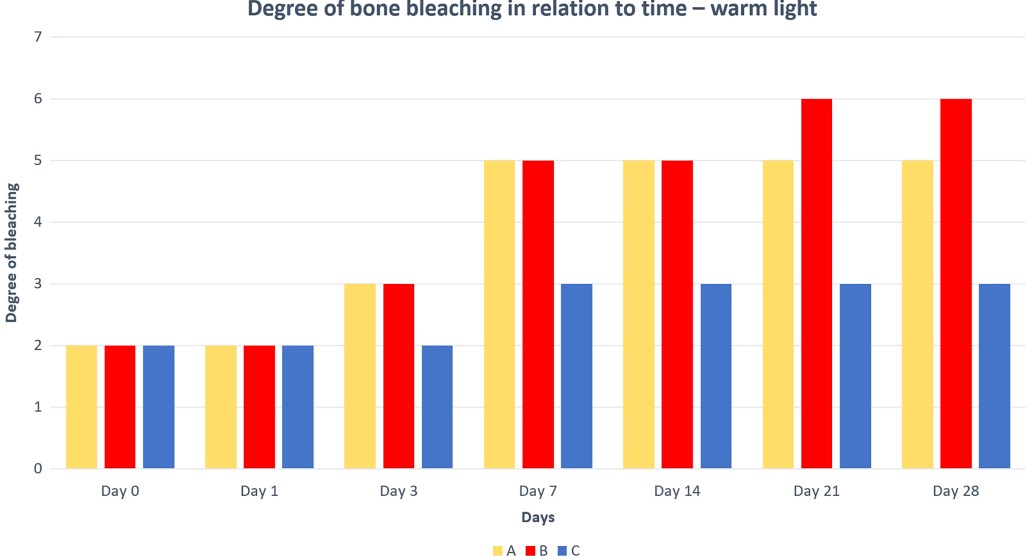

**Figure 5 Degree of bone bleaching in relation to time—warm light.** A—macerated bone, B—boiled bone, C—natural bone, x-axis = days, y-axis = degree of bleaching.

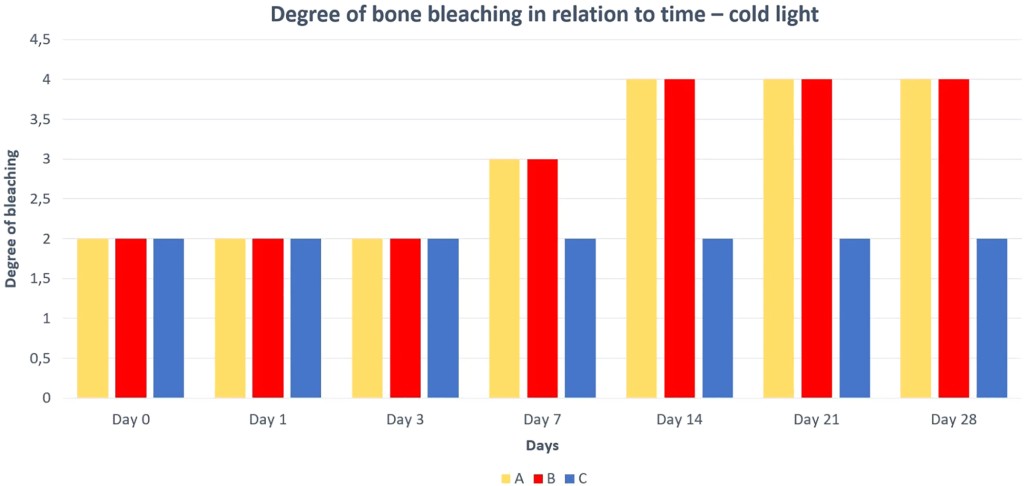

**Figure 6 Degree of bone bleaching in relation to time—cold light.** A—macerated bone, B—boiled bone, C—natural bone, x-axis = days, y-axis = degree of bleaching.

(*Gerritsen & Buschmann, 1960*). The use of physical methods (light and heat) is much safer than chemical ones. Fat is released from bone marrow adipocytes by heating and goes through a set of Haversian canals to the bone surface and further into the environment (*Adams, 1980*). The disadvantage is the lengthy process of degreasing, which can take several weeks depending on the ambient temperature. The use of sunlight to bleach bone material has been known for a very long time (*Stokes, Marquez-Grant & Greenwood, 2020*), but information of using artificial light sources is still insufficient in literature. The most common negative effect of bone exposure to the sun is confirmed by the results of our work. Due to weather conditions, especially temperature variations and occasional rain showers, minor erosions of the periosteum and compact cracks have been observed on the bones. However, these cracks made it easier to get fat from the bone marrow to the surface of the bone, where it was absorbed by the sand.

The significant difference in the speed and intensity of whitening of bones exposed to the sun and the other two methods correspond with the absence of UVA and UVB radiation in the artificial light sources used in our experiment. We believe that the use of full-spectrum incandescent bulbs (including the UV component) would improve the bleaching capacity of artificial lights. On the other hand, it would make the production process more expensive because the purchase price of these lights is high.

The bleaching effect of all three light sources was not sufficient due to the short exposure, but from a safety and economic point of view, it was a suitable alternative for common chemical methods.

## CONCLUSIONS

The results of our work confirmed that it is possible to use various light and heat sources for degreasing and bleaching of the bone.

The best bone degreasing was found for bones exposed to sunlight and the warm light of a 100 W bulb. The worst result was achieved with exposure to cold light environmentally friendly LED bulb. Bleaching with sunlight had the best effect, that achieved whiteness comparable to the use of chemical bleaches (such as hydrogen peroxide) after only 28 days of exposure. However, the bleaching of the bones exposed to both hot and cold light was probably insufficient due to short exposure or the absence of a UV component of the light spectrum.

## ACKNOWLEDGEMENTS

We would like thank the management of our institute and the university for providing, material and facility support. Without which we would not be able to create this work.

### Funding
The University of Veterinary Sciences Brno paid the APC for this article. The funders had no role in study design, data collection and analysis, decision to publish, or preparation of the manuscript.

### Grant Disclosures
The following grant information was disclosed by the authors:
The University of Veterinary Sciences Brno paid the APC for this article.

### Competing Interests
The authors declare that they have no competing interests.

### Author Contributions
- Ondřej Horák conceived and designed the experiments, performed the experiments, analyzed the data, prepared figures and/or tables, authored or reviewed drafts of the article, and approved the final draft.
- Martin Pyszko conceived and designed the experiments, performed the experiments, analyzed the data, prepared figures and/or tables, authored or reviewed drafts of the article, and approved the final draft.
- Václav Páral conceived and designed the experiments, analyzed the data, authored or reviewed drafts of the article, and approved the final draft.
- Ondřej Šandor performed the experiments, analyzed the data, prepared figures and/or tables, and approved the final draft.

### Data Availability
The raw data is available in Table 2.

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
