# Peer review of "Degreasing and bleaching bones using light sources as a tool to increase the safety of teaching osteology at the University of Veterinary Sciences Brno"

_PeerJ, doi:10.7717/peerj.14036_

## Round 0.1 · original submission · Major Revisions

Dear Authors,
The manuscript covers a method to be used worldwide by anatomists for preparation of osteological specimens. But it has to be improved technically and in language before acceptance. Please, do incorporate the suggestions given by our reviewers and if you do not agree with them then justify your statements properly. Our all reviewers are experts in their respective fields and their comments will surely help in making of a quality manuscript. I wish you all the best and expecting the revised manuscript asap.
Regards

Reviewer 1 ·

Basic reporting

There is no new information in this manuscript, all veterinary anatomists know these methods.

Experimental design

OK

Validity of the findings

OK

Reviewer 2 ·

Basic reporting

1.1. Language
26 “Fat loss” not “fat loosing”
29 Haversian canals
28-29 Alter “blood dye” to “haemoglobin staining” or equivalent
1.2. Literature Review
36-37 Poor grammar within sentence: should be “…within the practical exercises in anatomy at the University…”
45 no comma needed, should be “carcinogenicity and teratogenicity”
49 suggest “not always possible”
49 need alternative word from “gentle”
53 “femur bones” could be just “femurs”
57-58 poor grammar
55-57 needs better ethical statement
General comment – the entire document needs to be re-written since there are numerous grammatical errors of tense and word choice.
1.3. Structure
Good
1.4. Figures
No legends.
1.5. Raw data
Tables unclear.

Experimental design

2.3. Rigour
Appears to be open to error, for example when the lights were off, what was the remaining light level and could this change?
Needs better scorin, for example lines 109-110 what is “whiten completely” on the scale used by the experimenters?
What does “cross” mean in lines 86-89?
2.4. Methods
Needs better description of the different methods. Suggest a table to set out the methods against the test conditions. The groups need to be more logically named: Ac Af Bc Bf etc make no sense. Needs a colour chart for the 1-10 scale.

Validity of the findings

3.1. Impact
Would be useful if more evidence supplied. It would that this is more a description of possible methods rather than assessing the methods.
3.2. Robust analysis
No analysis performed. The time to reach different scale colours could be plotted easily. No reporting of scale colour changes over time – this would be easily represented in a table.
3.3. Conclusions valid and relevant
Lack of analysis, even a graphic, means that the conclusions are opinion and lack support of evidence.

Additional comments

The aim of the research is good, however it is poorly translated and contains areas in need of improvement as detailed above.

Reviewer 3 ·

Basic reporting

The basic idea behind the article is to bleach and degrease the bone by natural method to over come the adverse effect of chemical agents.
The suggestions for improvement:
1.line no.29 and 108 -use term Haversian canal instead of Havers canal
2.Key words: Write bone degreasing instead of bone preparation
2.In Table 1.Mention about the values 1,2,3,5,7,8,9 + etc..
3.Follow the guidelines for reference writing.

Experimental design

A .Material method requires detail explanation regarding procurement of bones from dead dogs under following points:
1. Were dogs preserved in 10 per cent formalin for teaching ???
or
2. Were dogs kept in deep freezer??
or
3.Used Immediately after autopsy ??
Because above points certainly affect the result.

Validity of the findings

No comment

Additional comments

Looking to the adverse effect of chemicals used for whitening and degreasing of bone, the findings of the article would be helpful for the anatomists involved in osteology teaching.

---

## Round 0.2 · Minor Revisions

Dear Authors,

As per the suggestions advised by our expert reviewers, please do the needful and revise as soon as possible.

Regards and good luck.

Reviewer 3 ·

Basic reporting

Clear and unambiguous and technically correct. Corrections done as mentioned in previous review.

Experimental design

No comment

Validity of the findings

The findings will be useful for Anatomist as a safe and suitable alternative to chemical methods of bleaching the bones.

Additional comments

Follow the uniform pattern of Reference writing, particularly as per guidelines of journal.

·

Basic reporting

I have read the prior reviews, and now the Rebuttal and revised MS.

Line 67 of Tracked changes use not "analyzes" but "analyses"
It would not be normal to capitalise Sun in this MS's context.
Please use "sex" of dogs and not "gender" as the latter does not apply.
Please recheck as the MS switches between British and American English. e.g. colour/color, etc.
Only 11 references are cited; is that all that exists in this field? It seems awfully few.
I did not see changes in reference format as reviewer requested, but I did not see problems, either.
The new Tables/Figures are a big improvement but "Tables" 3-5 are Figures (graphs), and should be changed as such. Also, they use basic Excel format-- I recommend fixing them so they are more polished and professional.
Line A in "Table 5" cannot always be seen and is different colour than in "Tables" 3-4- please fix this (at least note in caption where line A is when invisible).

Experimental design

OK

Validity of the findings

OK

---

## Round 0.3 · accepted · Accept

Dear Dr. Chen,

It is my pleasure to inform you that as per the recommendation of our expert reviewers, the manuscript "Degreasing and bleaching bones using light sources as a tool to increase the safety of teaching osteology at the University of Veterinary Sciences Brno" - has been Accepted for publication in PeerJ.

This is an editorial acceptance and you will be intimated for the list of further tasks before publication. So, I request you to be available for a few days to make the necessary things asap.

Regards and good luck with your future submissions.